# Parameters for Optoperforation-Induced Killing of Cancer Cells Using Gold Nanoparticles Functionalized With the C-terminal Fragment of *Clostridium Perfringens* Enterotoxin

**DOI:** 10.3390/ijms20174248

**Published:** 2019-08-30

**Authors:** Annegret Becker, Tina Lehrich, Stefan Kalies, Alexander Heisterkamp, Anaclet Ngezahayo

**Affiliations:** 1Department of Cell Physiology and Biophysics, Institute of Cell Biology and Biophysics, Leibniz University Hannover, 30419 Hannover, Germany; 2Institute of Quantum Optics, Leibniz University Hannover, 30167 Hannover, Germany; 3Lower Saxony Centre for Biomedical Engineering, Implant Research and Development, 30625 Hannover, Germany; 4Center for Systems Neurosciences (ZSN), Veterinary Medicine University Hannover, 30559 Hannover, Germany

**Keywords:** C-CPE, gold nanoparticle, GNOME-LP, apoptosis, necrosis

## Abstract

Recently, we used a recombinant produced C-terminus (D194-F319) of the *Clostridium perfringens* enterotoxin (C-CPE) to functionalize gold nanoparticles (AuNPs) for a subsequent specific killing of claudin expressing tumor cells using the gold nanoparticle-mediated laser perforation (GNOME-LP) technique. For a future in vivo application, it will be crucial to know the physical parameters and the biological mechanisms inducing cell death for a rational adaptation of the system to real time situation. Regarding the AuNP functionalization, we observed that a relationship of 2.5 × 10^−11^ AuNP/mL to 20 µg/mL C-CPE maximized the killing efficiency. Regardingphysical parameters, a laser fluence up to 30 mJ/cm^2^ increased the killing efficiency. Independent from the applied laser fluence, the maximal killing efficiency was achieved at a scanning velocity of 5 mm/s. In 3D matrigel culture system, the GNOME-LP/C-CPE-AuNP completely destroyed spheroids composed of Caco-2 cells and reduced OE-33 cell spheroid formation. At the biology level, GNOME-LP/C-CPE-AuNP-treated cells bound annexin V and showed reduced mitochondria activity. However, an increased caspase-3/7 activity in the cells was not found. Similarly, DNA analysis revealed no apoptosis-related DNA ladder. The results suggest that the GNOME-LP/C-CPE-AuNP treatment induced necrotic than apoptotic reaction in tumor cells.

## 1. Introduction

The gold nanoparticle-mediated (GNOME) laser perforation (GNOME-LP) technology was developed to achieve a gentle cell permeabilization allowing the entry of small molecule into cells while maintaining a maximal survival [1,2]. In this context, the GNOME-LP technique was used for an accurate and high-throughput analysis of gap junction coupling [3] and for an efficient knock down of genes using specific siRNAs or morpholinos [2].

In a previous work, we showed that a recombinant produced C-terminus (D194-F319) of the *Clostridium perfringens* enterotoxin (C-CPE) could be used to functionalize gold nanoparticles (AuNPs) for a subsequently optoperforation-induced killing of tumor cells expressing the CPE receptors, claudin-3, -4, and -7 using GNOME-LP [4]. The advantage of using the C-CPE polypeptide is the binding ability to claudins without causing cytotoxicity as compared to the full-length CPE [5]. Moreover, C-CPE-functionalized AuNPs (C-CPE-AuNPs) allowed specific targeting of cells and increased the killing efficiency [4]. In order to kill tumor cells, the applied laser fluence was increased while the scanning velocity was reduced in comparison to the experiments, in which a maximal cell survival was the objective. The mechanisms of cell killing are still a matter of speculation, hindering a rational optimization of the technique in terms of maximal ablation of tumor cells with minimal action on non-tumor tissue.

GNOME-LP affects cells by combined laser-induced heating due to the plasmon resonance, which can lead to formation of plasmonic bubble [2,6]. The presumable cause of cell death may be related to induced necrosis or apoptosis in the cells. Apoptosis, also called programed cell death, is a cellular reaction related to specific proteases called caspases that cleave cellular structure proteins, nuclear proteins, and DNA [6]. The resulting DNA fragments are integer multiples approximately 200 bp in size. Separated in gel electrophoresis, these DNA fragments produce a characteristic ladder pattern [7,8,9,10]. Apoptotic reactions also involve mitochondrial damage resulting in breakdown of the mitochondrial membrane potential (MMP) and ATP depletion [11]. The MMP breakdown results in reduction of mitochondrial staining using MitoTracker, allowing an analysis of apoptosis using fluorescence microscopy [12,13]. Moreover, the ATP depletion suppress the activity of ATPase enzymes such as flippases, which are responsible for the backhaul of lipids like phosphatidylserine lipids in the inner membrane leaflet, allowing the maintenance of a polarized cell membrane with respect to lipid composition of the leaflets [14]. An increased presence of phosphatidylserine lipids in outer leaflet as a result of induction of apoptosis can be demonstrated using annexin V staining and represents another marker of a cell undergoing apoptosis [15,16]. In tissues, apoptosis is a cellular dismantling with a relatively stable plasma membrane. The cells undergoing apoptosis are finally cleared by macrophages before lysis and release of intracellular molecules, thereby avoiding induction of an immune response [17]. In contrast, necrosis is unregulated or an accidental cell death, mostly induced by exogenous stress and correlated with cell lysis with release of intracellular material in the tissue. The release of intracellular material acts as chemoattractant and activates an inflammatory reaction [18]. In tumors, necrosis may allow release of tumor neoantigens that could act like microbial pathogens-associated molecular patterns (PAMPs) or damage-associated molecular patterns (DAMPs) and activate immune-specific responses [19,20,21,22,23]. Such tumor neoantigens are expressed by tumor cells. However, they are maintained in the cells and rarely presented to the immune cells due to loss of HLA class I by tumor cells [24,25,26]. By inducing necrosis in tumor, it may be possible to increase the presence of these tumor antigens in the extracellular space within the tumor. The immune cells recruited by necrosis will find the liberated tumor neoantigens in an accessible environment, rendering possible an induction of immunological response to the tumor cells [20].

Related to our recent study demonstrating optoperforation-mediated killing of tumor cells with C-CPE-AuNPs, the present report analyzed the physical parameters and the biological mechanisms inducing the cell death with the objective to determine the range of adaptation for future in vivo application for clinical therapeutic intervention.

## 2. Results

In a recent report, we showed that C-CPE -functionalized AuNPs in combination with GNOME-LP efficiently killed tumor cells expressing CLDN-3, -4, and -7, documented by the uptake of membrane impermeable of molecule such as propidium iodide [4]. In the present report, we analyzed further physical parameters that might affect the efficiency of the GNOME-LP-induced cell killing. Moreover, we determined whether apoptotic mechanisms are involved in the cell death.

Using Caco-2 cells, we analyzed whether changing the concentration of the C-CPE during functionalizing of the AuNPs could change the GNOME-LP-induced cell killing efficiency. AuNPs (2.5 × 10^10^ particle/mL) functionalized with 5 µg/mL C-CPE in combination with GNOME-LP at a laser fluence of 30 mJ/cm^2^ with scanning velocity of 5 mm/s revealed a cell survival of 40% (Figure 1A). Increasing the concentration of the C-CPE during the functionalization process to 12 µg/mL, 20 µg/mL, and 50 µg/mL reduced the cell survival to 12%, 8%, and 13%, respectively (Figure 1A), showing an increased cell killing efficiency compared to a concentration of 5 µg/mL C-CPE.

Next, GNOME-LP was performed using a laser fluence of 5 mJ/cm^2^ as well as 20 µg/mL and 50 µg/mL C-CPE for AuNP functionalization to evaluate whether the increased C-CPE concentration could allow further reduction of the laser fluence for efficient cell killing. The reduced laser fluence did not affect cell survival for both C-CPE concentrations used for AuNP functionalization (Figure 1B). Taken together, the increased C-CPE concentration from 20 µg/mL to 50 µg/mL for AuNP functionalization did not increase the efficiency of cell killing using GNOME-LP at a fluence of 5 mJ/cm^2^. After we observed that 20 µg/mL C-CPE coupled to Strep-Tactin Chromeo conjugate also saturated the binding of C-CPE onto the cells (Appendix A) and that 2.5 × 10^10^ AuNPs/mL were the more effective for cell killing (Appendix A), we chose to use 20 µg/mL C-CPE.

For functionalization of the AuNPs (2.5 × 10^10^ AuNPs/mL), in order to analyze the influence of other parameters on the efficiency of cell killing in this context, we analyzed how changes in laser fluence or scanning velocity affected the cell killing efficiency. Using a constant velocity of 5 mm/s we found that a laser fluence of 5 mJ/cm^2^ reduced the cell survival to 89% compared to the AuNP control group. Application of laser fluence to 10 mJ/cm^2^ correlated with a reduction of the cell survival to 29%. With a laser fluence of 20 mJ/cm^2^ we observed a cell survival of only 8%, which could not be further reduced by increased laser fluence to 30 mJ/cm^2^ or 60 mJ/cm^2^ (Figure 1 C), suggesting that a laser fluence of at least 20 mJ/cm^2^ was necessary for a maximal cell killing. Considering the scanning velocity, GNOME-LP at a laser fluence of 30 J/cm^2^ was applied at 5 mm/s, 10 mm/s, and 40 mm/s. The respective cell survival of 8%, 28%, and 46% (Figure 1D) indicated that a scanning velocity of 5 mm/s had the most effect on cell survival. Eventually, we applied GNOME-LP with a laser fluence of 30 mJ/cm^2^ at 5 mm/s in combination with functionalized AuNPs using 20 µg/mL C-CPE for further experiments.

Since tumors are 3D structures, we analyzed whether increasing the C-CPE concentration during the functionalization of the AuNPs increased the GNOME-LP-affected spheroid structure composed of Caco-2 or OE-33 cells (Figure 2). The spheroids were incubated for 3 h with the C-CPE-AuNPs before GNOME-LP with 60 mJ/cm^2^ and 5 mm/s. We compared images of the spheroids taken at a same position before and after three-time GNOME-LP treatments. The presence of the C-CPE-AuNPs reduced the number of intact Caco-2 spheroids compared to the spheroids incubated with non-functionalized AuNPs or untreated spheroids (Figure 2C). The OE-33 spheroids treated with the C-CPE-AuNPs showed almost intact spheroid structure after GNOME-LP but displayed 50% reduced spheroid area (Figure 2D). To assess the cell viability of the cells in Caco-2 or OE-33 spheroids after GNOME-LP treatment, the spheroids were stained with propidium iodide and calcein AM. In living cells, the non-fluorescent cell-permeant dye calcein AM is converted to green-fluorescent calcein. Z-stack images of the spheroids after GNOME-LP in the presence of the C-CPE-AuNPs showed large peripheral propidium iodide-positive cells (red) without green fluorescence surrounding a central core with calcein (green) positive cells without propidium iodide (Figure 2A,B). Compared to the OE-33 spheroids, Caco-2 spheroids showed a smaller core with calcein positive cells without propidium iodide. Confocal images 24 h after the GNOME-LP application showed further destruction of Caco-2 spheroids with less calcein-positive cells. OE-33 spheroids showed less propidium iodide positive cells in the peripheral regions (Appendix A). Further cultivation of Caco-2 spheroids and OE-33 spheroids until 72 h after GNOME-LP showed that in Caco-2 spheroids, the cells were not able to regrow and contained almost only propidium iodide positive cells and very low calcein-stained cells, indicating reduced cell survival (Appendix A).

### Caspase-3/-7 Independent Cell Death

The cell death can be achieved by apoptotic or necrotic reactions. Apoptosis is a cellular controlled reaction characterized by physiological changes like disruption of the mitochondria or inversion of the membrane exposing the phosphatidylserine lipids in the external leaflet of the cell membranes. These physiological changes are orchestrated by activation of specific proteases called caspases. To analyze whether the GNOME-LP-induced cell death was related to apoptotic reaction, we stained the mitochondria using MitoTracker™ Orange and the phosphatidylserine lipids with Atto-labeled annexin V 3 h and 24 h after GNOME-LP treatment at 10 J/cm^2^, 20 J/cm^2^, 30 J/cm^2^, and 60 J/cm^2^. MitoTracker labels mitochondria, with a mitochondrial membrane potential (MMP) below −60 mV [27]. An increased apoptosis correlates with disruption of the MMP, which results in a reduction of MitoTracker staining. Annexin V binds to the phosphatidylserine lipids, which are exposed during apoptotic cell death. The results showed a reduced mitochondrial staining in cells treated with GNOME-LP in the presence of C-CPE-functionalized AuNPs in comparison to non-functionalized AuNPs (Figure 3A). It is remarkable that the different laser fluence seemed to equally affect the mitochondria and the reduced mitochondria labeling lasted until 24 h after GNOME-LP. For the phosphatidylserine lipids, GNOME-LP of cells in the presence of functionalized AuNPs showed an increased annexin V staining (Figure 3B) compared to GNOME-LP of cells with non-functionalized AuNPs (Figure 2B). The quantification of the annexin V staining revealed that the fluorescence intensity of annexin V staining is increased at an applied laser fluence of 60 mJ/cm^2^ compared to the other applied laser fluences of 10 J/cm^2^, 20 J/cm^2^, and 30 J/cm^2^ (Figure 3C), suggesting a fluence-dependent increase in the possibility of annexin V to reach the phosphatidylserine lipids.

Apoptotic reactions are induced by activation of caspases that digest different structure proteins and induce DNA degradation in specific DNA lengths that can be separated in gel electrophoresis, where they form characteristic ladder pattern [28]. Within the cascade family, caspase 3 and 7 are known as executioner caspases. These caspases induce DNA digestion and when they are activated, the cell cannot be rescued anymore. We analyzed whether these caspases were activated by GNOME-LP application and whether the inhibition of these caspases could protect the cells from GNOME-LP treatment in presence of C-CPE functionalized AuNPs. As shown in Figure 4A and Appendix A, application of GNOME-LP in the presence of C-CPE-AuNPs did not increase the caspase activity. We found a tendency towards reduction. Moreover, the presence of 20 µM carbobenzoxy-valyl-alanyl-aspartyl-(O-methyl)-fluoromethylketone (Z-VAD-FMK), a broad spectrum caspase inhibitor, reduced the activity of caspase 3 and 7 but did not antagonize the reduction of cell survival observed in cells treated with GNOME-LP in the presence of C-CPE-AuNPs (Figure 4B, Appendix A). Additionally, the analysis of the DNA of cells at any time after application of GNOME-LP in combination with C-CPE-AuNPs as shown for 3 h and 24 h revealed no DNA ladder pattern, which is commonly associated with the activation of Caspase 3/7. The DNA smear (Figure 4C) combined with the observation that the caspase inhibitor Z-VAD-FMK could not stop the action of GNOME-LP and C-CPE-AuNPs on the cells (Figure 4B) suggest that the cell death is not induced by apoptotic reaction. It is more likely that the cell death was related to induction of necrosis.

In summary, the results confirm our previous finding showing that C-CPE can efficiently be used to functionalize AuNPs following optically induced elimination of tumor cells expressing claudins. Moreover, we showed that increasing the C-CPE concentration to 20 µg/mL for the functionalization of AuNPs producing C-CPE-AuNPs increased the killing effect while reducing the applied laser fluence. Note that all improvements of GNOME-LP-induced cell killing could not allow us to increase the applied velocity. For 3D cultivated tumor cells, C-CPE-AuNPs induced cell death to a different extent depending on the cell type. At the biological level, the killing effect correlated with reduced mitochondria activity and increased annexin V staining. However, the cell death caused by GNOME-LP applied in the presence of C-CPE-AuNPs induced no increased caspase activity, and inhibition of caspases with Z-VAD-FMK did not antagonize the killing effect.

## 3. Discussion

In a previous report, we showed that a combination of GNOME-LP- and C-CPE-functionalized AuNPs could specifically be used to kill claudin-expressing tumor cells [4]. In order to prepare future applicability of the technique in therapeutic intervention, it is necessary to understand the physical and cellular mechanisms that lead to cell disruption. For that, we analyzed how a maximal cell ablation in the shortest time could be achieved with the technique. Different parameters such as the functionalization of AuNPs in combination with purely physical parameters, like the laser fluence and the velocity of GNOME-LP application, were tested. With respect to the functionalization of the AuNPs, it can be assumed that the efficiency is related to the sum of particles attached to the cells, which is controlled by the available claudin proteins in the cells, and the C-CPE-AuNPs complex. Binding studies with different C-CPE concentrations applied to the cells indicated an elevated C-CPE binding with increasing concentration up to 20 µg/mL (Appendix A). With a molecular weight of 15–20 kDa for C-CPE, 20 µg/mL would correlate with a molar concentration of about 10^−6^ M. This is in agreement with the observation that the cells used in this report surely expressed claudin-3, -4, and -7 [4], which were shown to have a very low *K_d_* of 1.2 × 10^−8^ M, 9.1 × 10^−9^ M, and 1.1 × 10^−8^ M, respectively [29,30]. The complex formation is determined by the interaction of the N-terminal Strep-tag II of C-CPE and the Strep-Tactin conjugated to AuNPs (C-CPE-AuNP complex)**.** We produced the used AuNP–C-CPE complexes by adding Strep-tagged C-CPE to a solution of citrate-stabilized AuNP-Strep-Tactin complex (see materials and methods). The fabrication of AuNP–Strep-Tactin complexes was performed by Aurion (Aurion, Wageningen, Netherlands) by conjugating Strep-Tactin (IBA, Göttingen, Germany) to AuNPs (Aurion) according to the method described by Fens [31]. The method allows the production of non-flocculated AuNPs covered by a single layer of Strep-Tactin molecules [32]. Due to the high affinity between the Strep-tag (on C-CPE) and the Strep-Tactin (on the AuNP), C-CPE binds very well on the AuNP–Strep-Tactin complex [33], resulting in stable AuNP–C-CPE complex. If we assume a size of about 50 kDa for Strep-Tactin, we can estimate that each nanoparticle was able to react with 5000 Strep-Tactin molecules. We can, therefore, assume that with 20 µg/mL C-CPE, we generated a quantity of C-CPE-AuNPs complex, allowing us to target all cells used in the present reports, which expressed claudin-3, -4, and -7 with the highest affinity for CPE. For cells that might express low-affinity claudins, such as claudin-8 with *K_d_* ≈ 10^−6^ M or claudin-14 with *K_d_* ≈ 2.8 × 10^−6^ M [29,30] as well as for future clinical investigations in which tumors containing different cells should be expected, further adjustment of dosage could be necessary.

With respect to the physical parameters, we found that with laser fluence between 5 mJ/cm^2^ and 60 mJ/cm^2^, the killing efficiency increased, achieving a steady maximum at 20–30 mJ/cm^2^ (Figure 1C). The optimized C-CPE functionalization of AuNPs did not change the relationship between the laser fluence and the killing efficiency (Figure 1C; [4]). However, the AuNP functionalization increased the absolute killing efficiency observed for the different fluencies. Moreover, it was striking that independently of the applied laser fluence and for the functionalization process, the killing was maximal at a scanning velocity of 5 mm/s [4]. An application velocity of 5 mm/s and a laser fluence of 30 mJ/cm^2^ reduced the survival to 8%. The increase of velocity to 10 mm/s or 40 mm/s allowed a survival of 30% and 45%, respectively (Figure 1D). The importance of the scanning velocity may be related to the overlap degree (OD) of the laser pulses as given by OD = (1 − *V_s_*/(*f* × *d*)) × 100, with *V_s_* representing the scanning velocity in mm/s, *f* the pulse repetition rate in kHz, and *d* the diameter of the laser pulse in µm [34]. In our experiments, the scanning velocity of 5 mm/s, 10 mm/s, or 40 mm/s corresponded to OD of, respectively, 99.7%, 99.4%, and 97.5%. However, the killing efficiency was not linearly related to OD. Changes of OD from 99.7% to 97.5% correlated with an exponential reduction of killing efficiency from over 90% to the asymptotic value of less than 70% (Figure 1D). The reduction of the OD as a consequence of the increased scanning velocity may reduce the number of laser pulses per particle. Dependent on the density of particles on the cell surface, less particles might interact with the laser with higher velocities. Consequently, the accumulated extent of membrane damage might decrease with higher velocities, while the damage related to a single particle remained the same. This could also partly explain why we could increase the efficiency of killing with a scanning velocity of 40 mm/s by repetition of the scanning (Appendix A).

The non-linear relationship between the killing efficiency and the OD (Figure 1D) was also observed between the killing efficiency and the laser fluence (Figure 1C). The killing efficiency increased exponentially with increasing fluence to achieve an asymptotic value at a fluence of 20–30 mJ/cm^2^ (Figure 1C).

Taken together, the results show that to achieve a good cell ablation in a reasonable time, increasing the laser power will not be enough. The pulse repetition rate and the pulse diameter should be adjusted to achieve both a laser fluence of 20–30 mJ/cm^2^ and an OD of more than 99.7%.

At the biological level, cell death is related to different mechanisms like apoptosis or necrosis. Apoptosis is a cellular reaction related to activation of caspase cascades that cleave different cellular proteins culminating in disruption of the plasma membrane, mitochondria, and chromatin integrity [35]. These processes, which constitute the hallmark of apoptosis, can be demonstrated by staining the phosphatidylserine lipids of the membrane with annexin V and mitochondria with MitoTracker as well as DNA ladder analysis. Furthermore, the apoptosis can be inhibited by pharmacological inhibition of apoptosis [36].

In our experiments, we observed that application of GNOME-LP in combination with C-CPE-functionalized AuNPs induced an increase of annexin V staining (Figure 2B), while reducing the staining of the mitochondria (Figure 2A) suggesting that annexin was able to reach the phosphatidylserine lipids and that mitochondria were disrupted. The increased annexin V staining was reduced 24 h after optoperforation (60 mJ/cm^2^). It is possible that this reduction was related to the strongly reduced number of cells that were found at this time point (Figure 3C). In addition, nuclei staining showed nuclei shrinkage of GNOME-LP-treated cells in the presence of the C-CPE-AuNPs. The results can be interpreted as evidence for apoptosis. However, this assumption was not confirmed by the analysis of caspase 3/7 activity, which was not increased upon GNOME-LP application (Figure 4A and Appendix A). Moreover, the caspase inhibitor could not prevent the GNOME-LP/C-CPE-AuNPs-induced cell death (Figure 4B). To explain these contradictory results, we postulate that induced permeabilization of the cell membrane allowed propidium iodide as well as annexin V uptake (Appendix A). The strong permeabilization of the cell membrane caused mitochondria disruption, including loss of their membrane potential, which led to the low mitochondria staining (Figure 3A). Additionally, DNA ladder analysis 24 h after laser exposure revealed a slight smear rather than a DNA ladder pattern in gel electrophoresis (Figure 4C). The results suggest that the combination of GNOME-LP/C-CPE-AuNPs might have killed the cells by induction of necrotic reactions.

In 3D system, the killing efficiency was reduced (Figure 2) compared to cells cultivated in 2D system. Even after three times application of GNOME-LP the killing efficiency for cells cultivated in 3D was reduced compared to 2D cultivation. It is possible that the penetration of the laser in the spheroids was an issue. Usage of near-infrared laser with adapted AuNPs would probably ameliorate the results. Moreover, we found that the type of cell forming the spheroids played a role (Figure 3), since we observed differences between spheroids that formed Caco-2 and OE-33-cells. The reason for the differences is not clear; it could be related to the difference in origin of the cells. Nevertheless, the data show that optoperforation in combination with C-CPE-functionalized AuNPs is applicable to 3D structures and, thus, could constitute an alternative tool for tumor treatment.

In contrast to apoptotic reactions, necrotic reactions strongly activate immunological reaction in tissues [37]. With respect to tumor treatment, this could be an advantage. Necrotic reaction in tumor could enable the attraction of immunological cells into the tumors, which will attack and destroy the tumor, thus increasing the treatment efficiency. Methods to recruit immunological cells are discussed in tumor therapy. Some reports show that manipulating the tumor environment could induce immune cells to attack the tumor [19,21,22,23]. Other reports show that tumor cells express neoantigens, but these antigens are not present on the surface of the tumor cells due to loss of HLA class I by tumor cells [24,25,26]. By inducing the necrosis, it may be possible to liberate these tumor antigens in the tumor interstitial spaces. The immune cells recruited by the necrosis could target the locally liberated tumor neoantigens and mount an immunological response to the tumor cells.

## 4. Materials and Methods

### 4.1. C-terimal Fragment of Clostridium Perfringens Enterotoxin (C-CPE)

The C-CPE_194–319_ protein with an N-terminal Strep-tag II was prepared as previously described [4] and stored at 4 °C in the presence of 0.02% sodium azide as preservative.

### 4.2. Cell Culture

Human tumor cell lines Caco-2 (DSMZ, Braunschweig, Germany) and OE-33, (provided by Dr. Bianca Nitzsche, Charité - Universitätsmedizin Berlin) were grown in tissue-treated petri dishes at 37 °C in a humidified atmosphere containing 5% CO_2_. Caco-2 cells were cultured in DMEM Ham’s F12 (FG 4815, Biochrom, Berlin, Germany) medium and OE-33 cell were cultured in RPMI (FG 1215, Biochrom) medium. Bot culture media were supplemented with 10% fetal calf serum (Biochrom), 100 units/mL penicillin (Biochrom), and 100 µg/mL streptomycin (Biochrom).

For spheroid formation, Caco-2 cells and OE-33 cells were grown in growth factor reduced Matrigel (354230, Corning) as previously described [4]. In brief, 60 µL of Matrigel was added to the growth surface of µ-Plate 96 well (ibidi, Martinsried, Germany) and placed in cell culture incubator for 30 min to allow gel formation. The Matrigel was overlaid with cell culture medium and 5 × 10^4^ cells were added. The cells were cultured for 3–4 days to allow spheroid formation.

### 4.3. Gold Nanoparticle-Mediated Laser Perforation (GNOME-LP)

GNOME-LP was performed as previously described [4] using laser HLX-G-F020-10103 (Horus Laser S.A.S, Limoges, France). The laser had the following characteristics: Pulse repetition rate 2.25 kHz, maximal power 100 mW, pulse diameter 80 µm [2]. To generate the C-CPE-AuNP complex, we used AuNP (Ø: 25 nm; Aurion) conjugated with Strep-Tactin (IBA). The conjugation of Strep-Tactin to the AuNPs was performed by Aurion according to the method described by [31]. The procedure was developed to allow a production of non-clustered AuNP-Strep-Tactin complexes with a single protein layer [32]. To produce the C-CPE-AuNP complexes used in the experiments, C-CPE and AuNPs were mixed and incubated overnight at 4 °C to allow the binding of Strep-tag II (on C-CPE) and the Strep-Tactin (on the AuNPs). Cell culture medium was added to achieve the reported C-CPE concentration (Figure 1A) and a constant AuNP-Strep-Tactin concentration of 2.4 × 10^10^ AuNP-Strep-Tactin/mL

For GNOME-LP, Caco-2 cells pre-incubated with the C-CPE-AuNP complex were exposed to the pulsed laser with 30 mW (30 mJ/cm^2^) at 5 mm/s. After optical treatment, propidium iodide (10 µM) and Hoechst 33258 (1 µM) were added to the cells for 30 min. Imaging of propidium iodide (excited at 535) and Hoechst (excited at 346 nm) uptake was performed with a Nikon Ti-E inverted fluorescence microscope (Nikon, Düsseldorf, Germany) using 4× objective and Nikon Software Nis-Elements 4.4 (Nikon, Düsseldorf, Germany). Dead cells were indicated by propidium iodide and Hoechst uptake, while healthy cells were only Hoechst positive. Propidium iodide and Hoechst positive cells were counted using ImageJ/Fiji. The cell survival is indicated by the differences between the number of cells stained by Hoechst and the number of the cells that were stained by both Hoechst and propidium iodide.

GNOME-LP of Caco-2 spheroids and OE-33 spheroids were pre-incubated with the C-CPE-AuNPs. Transmission images of the spheroids were taken in the same plane and at the same position before and after GNOME-LP (60 mJ/cm^2^; 5 mm/s) using 10× objective of the Ti-E inverted fluorescence microscope (Nikon). The spheroids were evaluated by comparing the intact spheroids or the spheroid area before and after GNOME-LP treatment using ImageJ/Fiji.

### 4.4. Activity of Caspase-3 /-7

The activity of caspase-3 /-7 in Caco-2 cells was measured by using the Caspase-Glo^®^ 3/7 Assay (Promega, Mannheim, Germany) according to the manufacturer’s protocol. In brief, 100 µL Caspase-Glo^®^ 3/7 reagent was added to each well containing GNOME-LP-treated cells in 100 µL cell culture medium. Contents were mixed using plate shaker at 160 rpm for 30 s and incubated at room temperature for 1 h. Luminescence of each sample was measured in Mithras LB 940 plate reader (Berthold, Bad Wildbad, Germany). For caspase inhibition cells were treated with 20 µM Z-VAD-FMK (Promega, Mannheim, Germany), a caspase family inhibitor, 30 min prior GNOME-LP.

### 4.5. DNA Ladder

For detection of DNA ladder, the Apoptotic DNA Ladder Kit (11835246001, Sigma-Aldrich, Taufkirch, Germany) was used according the manufacturer’s protocol. The purified DNA (100 ng) was separated in 1% agarose gel stained with GelRed (Biotium, Hayward, CA, USA).

### 4.6. Detection of Cell Death

Phosphatidylserine lipids of GNOME-LP-treated Caco-2 cells were stained with annexin V Atto488 conjugate (ALX-209-258-T100, Enzo Life Science) diluted in cell culture medium (1:40) for 15 min at room temperature. Mitochondria of GNOME-LP-treated Caco-2 cells were stained with 50 ng MitoTracker™ Orange (Thermo Fisher Scientific, Waltham, MA, USA) in cell culture medium for 30 min in cell incubator. Cell nuclei were stained with Hoechst (1 µM). For evaluating cell death in 3D matrix, spheroids were stained with propidium iodide (10 µM) and calcein AM (1:1000) diluted in serum-free cell culture medium for 30 min in cell culture incubator.

Fluorescence images were captured on Nikon Eclipse TE2000-E confocal laser scanning microscope (488 nm for annexin V Atto488 or calcein AM, 535 nm for MitoTracker™ Orange, 461 nm for Hoechst, Nikon, Düsseldorf, Germany) using 60× water immersion objective and EZ-C1 3.80 software program (Nikon, Düsseldorf, Germany). The grade of phosphatidylserine lipids exposure was estimated by normalization of the annexin V Atto488 fluorescence intensity to the nuclei count. Fluorescence images were processed with the ImageJ/Fiji software.

### 4.7. Statistical Analysis

Data are presented as mean ± SEM. Statistical comparison between groups was performed using the Student’s paired two-sided *t* test. *p* < 0.05 was considered to be significant.

## Figures and Tables

**Figure 1 ijms-20-04248-f001:**
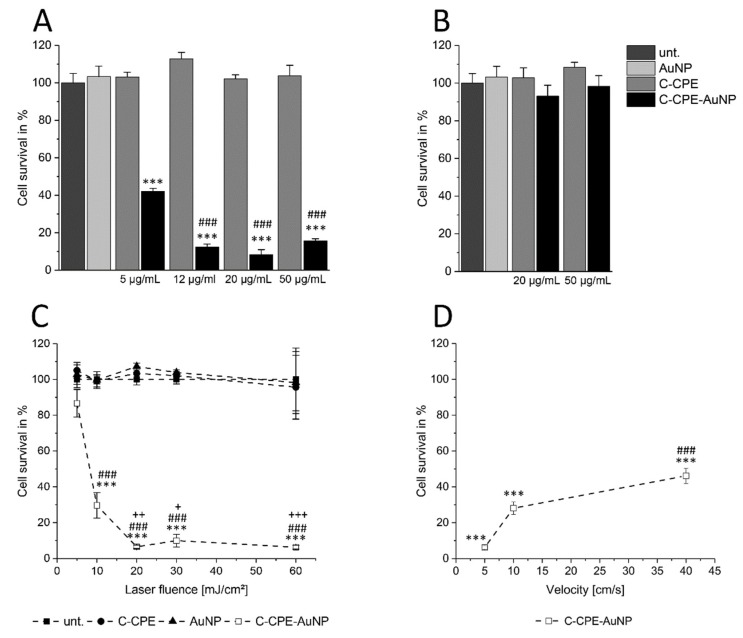
Increased C-CPE concentration for AuNP functionalization increased the efficiency of gold nanoparticle-mediated laser perforation (GNOME-LP)-induced cell killing of Caco-2 cells. (**A**) The survival of Caco-2 cells evaluated by propidium iodide uptake in cells treated with GNOME-LP (30 mJ/cm^2^ at 5 mm/s) in presence of C-CPE, AuNPs, and C-CPE functionalized AuNPs (C-CPE-AuNPs). The increase of C-CPE concentration during functionalization process reduced the cell survival. The results were analyzed with Student’s *t* test. * Significant difference to the AuNP control group: *** *p* < 0.001. # Significant difference to 5 µg/mL C-CPE functionalized AuNPs: ### *p* < 0.001. (**B**) GNOME-LP cell killing with lower laser fluence (5 mJ/cm^2^ at 5 mm/s) in combination with AuNPs functionalized with 20 µg/mL and 50 µg/mL C-CPE did not affect the cell survival. (**C**) Killing efficiency of AuNP functionalized with 20 µg/mL C-CPE and GNOME-LP (30 mJ/cm^2^) applied at indicated scanning velocity. The results were analyzed with Student’s *t* test. * Significant difference to the AuNP control group: *** *p* < 0.001. # Significant difference to velocity of 5 mm/s: ### *p* < 0.001. (**D**) GNOME-LP with different laser fluence (5–60 mJ/m^2^; 0.5 s/cm) in combination with 20 µg/mL C-CPE functionalized AuNPs. The results were analyzed with Student’s *t* test. * Significant difference to the AuNP control group: *** *p* < 0.001. # Significant difference to laser energy of 5 mJ/cm^2^: ### *p* < 0.001. + Significant difference to the laser fluence of 10 mJ/cm^2^: + *p* < 0.05, ++ *p* < 0.01, +++ *p* < 0.001. All graphs represent the mean ± SEM of cell survival relative to untreated cells (unt.).

**Figure 2 ijms-20-04248-f002:**
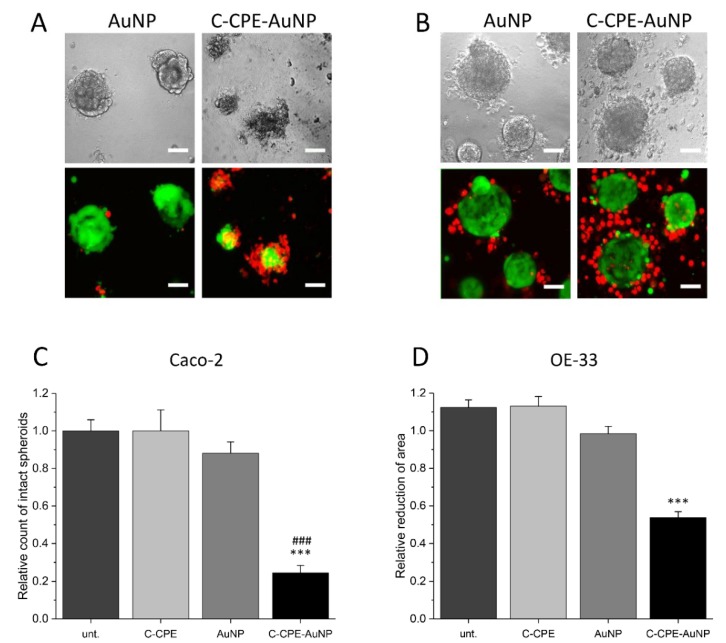
Caco-2 and OE-33 spheroids after three-time application of GNOME-LP. (**A**,**B**) Confocal images of Caco-2 (A) and OE-33 (B) spheroids treated with GNOME-LP in the presence of AuNPs or C-CPE-AuNPs. C-CPE-AuNPs destroyed Caco-2 spheroid structure and reduced cell survival as shown by the large amount of propidium iodide (red)-positive cells and the reduced amount of calcein-positive cells. For OE-33 spheroids, treatment with GNOME-LP in the presence of C-CPE-AuNPs induced cell death (propidium iodide-positive cells) in the peripheral regions. The cells in the core were still calcein-positive and propidium iodide-negative, indicating that they were still alive. Scale bar, 50 µm. (**C**) Quantification of intact Caco-2 spheroids indicating reduced intact spheroids after GNOME-LP with C-CPE-AuNPs. Graphs represent the mean ± SEM of intact spheroids count relative to spheroids before GNOME-LP application. The results were analyzed with Student’s *t* test. * Significant difference to untreated spheroids: *** *p* < 0.001 and ### *p* < 0.001 significant difference to the AuNPs treated spheroids. (**D**) Quantitative evaluation of OE-33 spheroid area showing that GNOME-LP applied in the presence of C-CPE-AuNPs reduced the spheroid area compared to the spheroid treated in absence of C-CPE-AuNPs. Graphs represent the mean ± SEM of the spheroid area reduction relative to spheroid area before laser irradiation. The results were analyzed with Student’s *t* test. * Significant difference to the spheroid area before laser irradiation: *** *p* < 0.001.

**Figure 3 ijms-20-04248-f003:**
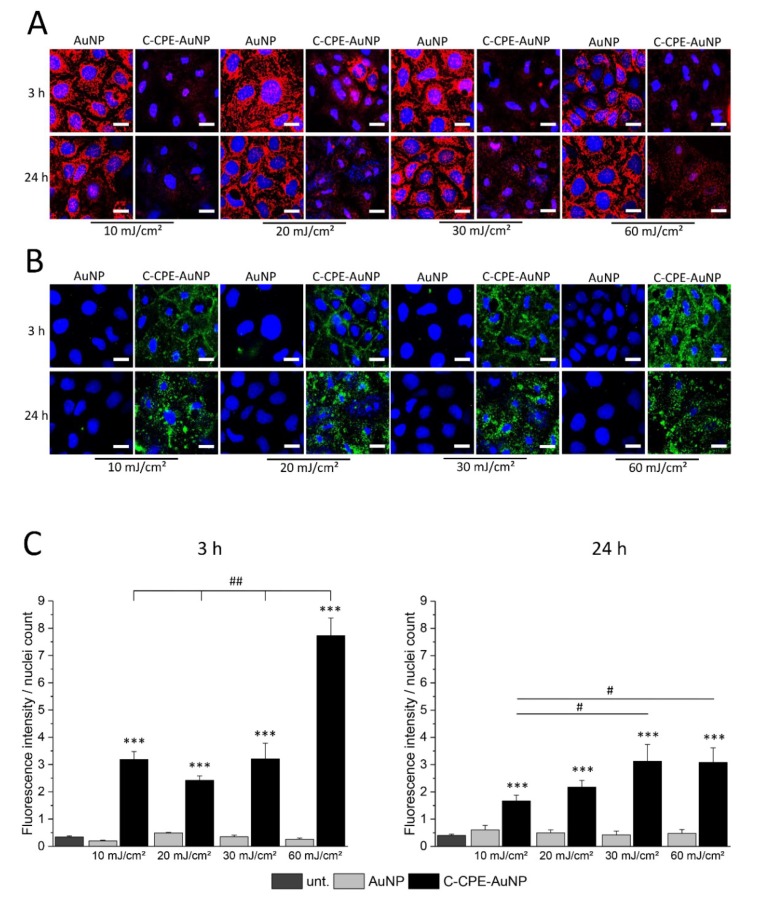
GNOME-LP in presence of C-CPE-AuNPs affected the annexin V and MitoTracker staining of the cells. (**A**) Independent of the applied laser fluence, 3 h and 24 h after GNOME-LP treatment in the presence of C-CPE-AuNPs reduced MitoTracker™ Orange mitochondria labeling (red) and (**B**) induced annexin V Atto488 staining of cells (green). Remark the shrinkage of the nuclei (blue) in cells treated with GNOME-LP in presence of C-CPE-AuNPs. Nuclei were stained with Hoechst. Scale bar, 20 µm. (**C**) Quantification of the experiments showing the increased annexin V Atto488 staining of GNOME-LP-treated cells in the presence of C-CPE-AuNPs 3 h and 24 h after GNOME-LP treatment. The graphs represent the mean ± SEM of annexin V Atto488 fluorescence intensity relative to number of the nuclei. The results were analyzed with Student’s *t* test. * Significant difference to the untreated control group: *** *p* < 0.001. # Significant difference to other laser fluencies: ## *p* < 0.01, # *p* < 0.05.

**Figure 4 ijms-20-04248-f004:**
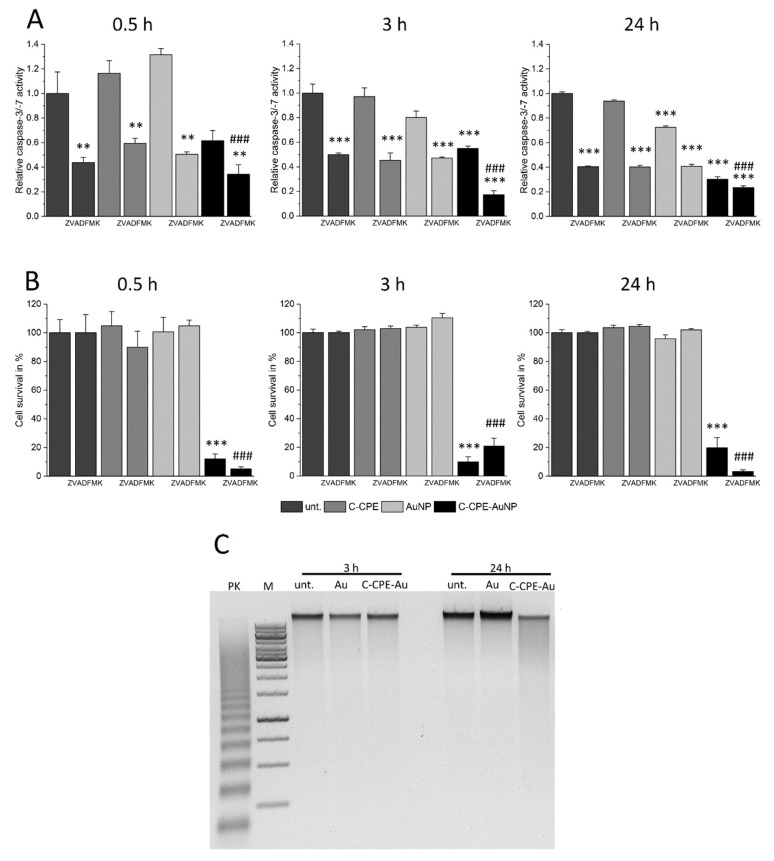
The caspase inhibition did not antagonize the cell death induced by GNOME-LP applied in presence of C-CPE-AuNPs. (**A**) Application of GNOME-LP (5 mm/s; 30 mJ/cm^2^) in the presence or absence of C-CPE-AuNPs did not increase the activity of caspase-3/-7. The presence of the caspase inhibitor Z-VAD-FMK (20 µM) was able to reduce the activity of caspase-3/-7. The graphs represent the mean ± SEM of caspase-3/-7 activity relative to untreated cells without Z-VAD-FMK as control reference. The results were analyzed with Student’s *t* test. * Significant difference to the control reference: ** *p* < 0.01, *** *p* < 0.001. # Significant difference to the control reference with Z-VAD-FMK: ### *p* < 0.001. (**B**) Applied in presence of C-CPE-AuNPs, GNOME-LP reduced the cell survival 0.5 h, 3 h, or 24 h after GNOME-LP application. The presence of Z-VAD-FMK (20 µM) was not able to protect the cells. The graphs represent the mean ± SEM of cell survival relative to untreated cells. The data were compared by Student’s *t* test. * Significant difference to the control reference: *** *p* < 0.001. # Significant difference to untreated cells with Z-VAD-FMK: ### *p* < 0.001. (**C**) Analysis of the DNA by gel electrophoresis showing that application of GNOME-LP in the presence of C-CPE-AuNPs did not induce DNA laddering; 100 ng DNA of each probe was applied to 1% agarose gel. Positive control (PK) DNA of U937 cells treated with 4 µg/mL camptothecin for 3 h (provided by the DNA ladder kit), 1 kbp marker (M), and DNA isolated from Caco-2 cells after 3 h and 24 h of laser irradiation.

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
