# Peer review of "Parameters for Optoperforation-Induced Killing of Cancer Cells Using Gold Nanoparticles Functionalized With the C-terminal Fragment of *Clostridium Perfringens* Enterotoxin"

_ijms, 2019, doi:10.3390/ijms20174248_

Round 1
Reviewer 1 Report
In my opinion the work entitled „ Parameters for optoperforation induced killing of cancer cells using gold nanoparticles functionalized with the C-terminal fragment of Clostridium perfringens enterotoxin” presents interesting and valuable results of studies. The manuscript is well-constructed and clearly written. Nevertheless, I think that the captions are too long and misleading for quick recognition of the results.
The scientific background and aims of the work were well-described in the part Introduction. The selection of the cell lines as well as the selection of experimental methods is proper and clear. Similarly, reading the part Discussion one can notice that the interpretation of gained results is well-presented and confronted with other available scientific reports.
I am convinced that the work should be published however I would like to advice improvements of some chemical issues. Generally, in my opinion the process of the functionalization of gold nanoparticles by enterotoxin is poorly described and in my opinion it is not reproducible. I tried to found more information about this process in the previous work of the authors (Sci. Rep. 2018), however besides the information about the size of nanoparticles and the description of mixing procedure, I did not gained information about other physicochemical properties of gold nanoparticles. It is worth mentioning that although the gold nanoparticles are considered as biocompatible material, many literature reports show that some specific molecules forming stabilizing layer around nanoparticles can enhance their toxicity. The authors obtained these nanoparticles from the company Aurion, Wageningen, Netherlands) and they should have described their stabilizers (maybe Citrate anions or PVP) and zeta potetential (negatively or positively charged). In my opinion these parameters may induce the interactions between the nanoparticles and enterotoxi. It is not clear if the enterotoxin is physically or covalently adsorbed on gold nanoparticles or maybe these particles do not interact each other. If possible, I suggest to determine zeta potential of unmodified gold nanoparticles and gold nanoparticles after their mixing with C-CPE (of course for the given ionic strength and pH) and to answer on this question. I would lie to mention also that the authors should express the concentration of gold nanoparticles as mass concentration. Then, it will be comparable with the concentration of applied C-CPE.
The abbreviation SEM is applied for description of scanning electron microscopy and in my opinion should not be used for standard error of the mean.
Author Response
The reviewer#II sent us a copy of our manuscript with recommended changes. We have considered the proposed changes along the manuscript. The changes are clearly recognizable in the marked version of the manuscript.
Review#01
Comments and Suggestions for Authors
In my opinion the work entitled „ Parameters for optoperforation induced killing of cancer cells using gold nanoparticles functionalized with the C-terminal fragment of Clostridium perfringens enterotoxin” presents interesting and valuable results of studies. The manuscript is well-constructed and clearly written. Nevertheless, I think that the captions are too long and misleading for quick recognition of the results.
The scientific background and aims of the work were well-described in the part Introduction. The selection of the cell lines as well as the selection of experimental methods is proper and clear. Similarly, reading the part Discussion one can notice that the interpretation of gained results is well-presented and confronted with other available scientific reports.
I am convinced that the work should be published however I would like to advice improvements of some chemical issues. Generally, in my opinion the process of the functionalization of gold nanoparticles by enterotoxin is poorly described and in my opinion it is not reproducible. I tried to found more information about this process in the previous work of the authors (Sci. Rep. 2018), however besides the information about the size of nanoparticles and the description of mixing procedure, I did not gained information about other physicochemical properties of gold nanoparticles. It is worth mentioning that although the gold nanoparticles are considered as biocompatible material, many literature reports show that some specific molecules forming stabilizing layer around nanoparticles can enhance their toxicity. The authors obtained these nanoparticles from the company Aurion, Wageningen, Netherlands) and they should have described their stabilizers (maybe Citrate anions or PVP) and zeta potential (negatively or positively charged). In my opinion these parameters may induce the interactions between the nanoparticles and enterotoxin. It is not clear if the enterotoxin is physically or covalently adsorbed on gold nanoparticles or maybe these particles do not interact each other. If possible, I suggest to determine zeta potential of unmodified gold nanoparticles and gold nanoparticles after their mixing with C-CPE (of course for the given ionic strength and pH) and to answer on this question. I would lie to mention also that the authors should express the concentration of gold nanoparticles as mass concentration. Then, it will be comparable with the concentration of applied C-CPE.
The abbreviation SEM is applied for description of scanning electron microscopy and in my opinion should not be used for standard error of the mean.
Answer
We thank the reviewer for the comments.
We have addressed the remarks of reviewer. The interaction between C-CPE and the AuNP to generate the C-CPE-AuNP complexes is achieved by interaction between the Strep-Tag on the C-CPE and the Strep-tactin conjugated to the AuNPs. This interaction is not a covalent binding, however the binding between Strep-Tag and Strep-Tactin one of the most efficient binding in the biochemistry (Schmidt et al 2007). The conjugation of Strep-Tactin to the AuNPs was performed by Aurion according to a protocol established by Frens (1973) that allows to generate non-clustered AuNPs with a single layer of protein.
In consideration of this response, we have changed the text as follows:
Line 268-276
We produced the used AuNP-C-CPE complexes by adding Strep-tagged C-CPE to a solution of citrate-stabilized AuNP-Strep-Tactin complex (s. Materials and methods). The fabrication of AuNP-Strep-Tactin complexes was performed by Aurion (Aurion, Wageningen, Netherlands) by conjugating Strep-Tactin (IBA, Göttingen, Germany) to AuNPs (Aurion) according to the method described by Fens [31]. The method allows the production of non-flocculated AuNPs covered by a single layer of Strep-tactin molecule[1]. Due to the high affinity between the Strep-Tag (on C-CPE) and the Strep-Tactin (on the AuNP), C-CPE binds very well on the AuNP-Strep-Tactin complex [32], resulting in stable AuNP-C-CPE complexes.
Line 372-382
GNOME-LP was performed as previously described [4] using laser HLX-G-F020-10103 (Horus Laser S.A.S, Limoges, France). The laser had following characteristics: pulse repetition rate 2.25 kHz, maximal power 100 mW, pulse diameter 80 µm [2]. To generate the C-CPE-AuNP complex we used AuNP (Ø: 25 nm; Aurion) conjugated with Strep-Tactin (IBA). The conjugation of Strep-Tactin to the AuNPs was performed by Aurion according to the method described by [31]. The procedure was developed to allow a production of non-clustered AuNP-Strep-Tactin complexes with a single protein layer[2]. To produce the C-CPE-AuNP complexes used in the experiments, Strep-tagged C-CPE was mixed with AuNP-Strep-Tactin complex and cell culture medium was added to achieve the reported C-CPE concentrations (Fig. 1A) and a constant AuNP-Strep-Tactin concentration of 2.5*1010 AuNP-Strep-Tactin /ml. The mixture was maintained at room temperature for 60 min to allow the binding between the Strep-tag (on the C-CPE) and the Strep-Tactin (on the AuNP).
Frens (1973) and Schmidt et al 2007 were added to the references as respectively (31) and (32).
[1] van de Plas, P. Willems, S.; Leunissen J. (2019) Gold nanoparticle conjugation adsorption or covalent binding? Aurion Newsletter 6
[2] van de Plas, P. Willems, S.; Leunissen J. (2019) Gold nanoparticle conjugation adsorption or covalent binding? Aurion Newsletter 6
Reviewer 2 Report
You need to proofread the manuscript, paying attention to "commas, etc.".
Please see the attached for further comments and recommendations

Author Response
The reviewer#II sent us a copy of our manuscript with recommended changes. We have considered the proposed changes along the manuscript. The changes are clearly recognizable in the marked version of the manuscript.